



# Inferring the source regions of pulsating auroras

Eric Grono[1] and Eric Donovan[1]

[1]Department of Physics and Astronomy, University of Calgary, Calgary, Alberta, Canada

**Correspondence:** Eric Grono (emgrono@ucalgary.ca)

**Abstract.** The aurora is an essential tool for remote sensing the large-scale dynamics of the magnetosphere which are to difficult to observe in situ and impossible to recreate in a lab. Despite pulsating aurora being a common and widespread morning-sector phenomenon, the processes responsible for its differentiation are not understood. In situ measurements of the pulsating aurora source regions are difficult to associate with specific auroral features, yet such observations will be necessary for identifying the unique causes of pulsating auroras. This study reports a method of inferring when a spacecraft is passing through the source region of patchy aurora (PA) based on the structuring of chorus. The locations of longer-lived chorus packets are found to correspond to the region of PA occurrence reported by Grono and Donovan (2019b). This result constrains the region where the structuring mechanisms and conditions responsible for PA and patchy pulsating aurora (PPA) can exist.

## 1 Introduction

Particle precipitation is the immediate cause of the aurora. Precipitating electrons and protons collide with neutral atmospheric particles to produce auroral emissions through collisional excitation and charge exchange. Contrary to the popular belief, the overwhelming majority of these particles do not come from the solar wind, but from within the magnetosphere, which is filled with magnetically trapped electrons and ions. These magnetically trapped particles will bounce between the north and south hemispheres, reversing direction at mirror points where the magnetic force causes them to change direction. The angle between a particle's velocity and the local magnetic field is called the *pitch angle*. At any point in the magnetosphere there is a range of pitch angles that result in particles mirroring at low enough altitudes that they will precipitate into the atmosphere and produce aurora, thereby escaping their trapping. This range of pitch angles is known as the *loss cone*.

Energy input by the solar wind drives dynamic plasma processes throughout the magnetosphere, some of which can modify particle pitch angles at a particular point in space and drive them into the loss cone. Once they are in the loss cone, their mirror point is at a low enough altitude that they can undergo collisions and cause aurora. If the magnetosphere was static, the altitude of particles' mirror points would be fixed, and they would remain stably bounce-trapped. Under these circumstances, all aurora that could ever be produced would occur on the first bounce.





The type of mechanism that drives particle precipitation is one basis on which the aurora can be categorized. *Discrete* auroras
form when electric fields parallel to the magnetic field accelerate charged particles and increase their kinetic energies parallel
to the magnetic field, thereby shifting their pitch angle into the loss cone. *Diffuse* auroras are created when charged particles
have their pitch angles scattered into the loss cone by stochastic processes, such as wave-particle interactions and magnetic
field curvature.

Pulsating aurora is a common type of diffuse aurora which is pervasive in the morning-sector auroral oval (Jones et al.,
2011; Partamies et al., 2017; Grono and Donovan, 2019b). Pulsating auroras can have diverse characteristics, but typically
have an irregular, patchy structure (Royrvik and Davis, 1977) that evolves in time (Shiokawa et al., 2010). These structures
are characterized by quasi-periodic pulsations and precipitating electrons with kinetic energies between a few to 100s of keV
(Johnstone, 1978).

Samara and Michell (2010) reported that auroral pulsations occur in two main frequency bands. Larger structures, 10s of km
across, that are frequently imaged by all-sky cameras typically have lower frequency pulsations between 50–500 mHz. Smaller
features, which are visible in narrow field of view imagers, can pulsate at higher frequencies between 0.5–15 Hz, although
frequencies as high as 54 Hz have been observed (Kataoka et al., 2012).

It is unknown for how long pulsating aurora events can persist, but ground-based optical observations see them last for
1.5 hours on average (Jones et al., 2011; Partamies et al., 2017). They can be much longer lived, however; Jones et al. (2013)
reported an event continuing for upwards of 15 hours. Pulsating aurora events frequently continue until sunrise when it is
no longer possible to image them, making measurements of their durations inherently conservative (Partamies et al., 2017).
The individual pulsating features that comprise an event have lifetimes ranging from a few seconds to 10s of minutes, largely
dependant on the type of pulsating aurora (Grono et al., 2017; Grono and Donovan, 2018).

Grono and Donovan (2018) subcategorized pulsating auroras into *amorphous pulsating aurora* (APA), *patchy pulsating
aurora* (PPA), and *patchy aurora* (PA). These types are differentiated based on the extent of structuring and pulsation they
exhibit. APA is a rapidly evolving form of pulsating aurora which is generally unstructured and can evolve so quickly it is
often impossible to uniquely identify specific features in successive images captured at a 3 second cadence. APA is the most
common type of pulsating aurora and it is nearly ubiquitous from 3 to 6 MLT (Grono and Donovan, 2019b). PPA and PA
are characterized by long-lived structures which can persist for 10s of minutes, but PPA pulsates while PA does not. Despite
this, due to the overall similarity between PA and PPA and how it often coexists with APA, PA is regarded as a non-pulsating
pulsating aurora (Grono and Donovan, 2018, 2019a, b), as oxymoronic as that is. From this point onward, the individual types
will be referred to by their acronyms and "pulsating aurora" will be used to refer to them collectively.

It is widely understood that pulsating aurora arises from resonant interactions between equatorial electrons and plasma waves
(e.g., Coroniti and Kennel, 1970; Davidson, 1990; Miyoshi et al., 2010). These interactions scatter some electrons into the loss
cone, producing electron precipitation and aurora. The shape of the pulsating auroral feature reflects the structuring of the
coherent wave-particle interactions in its source region.

Processes and conditions that modulate the effectiveness of wave-particle interactions are responsible for the existence of
bright "on" states and dim "off" states. The off states correspond to times when relatively few equatorial electrons are resonating





# How waves cause pulsating aurora

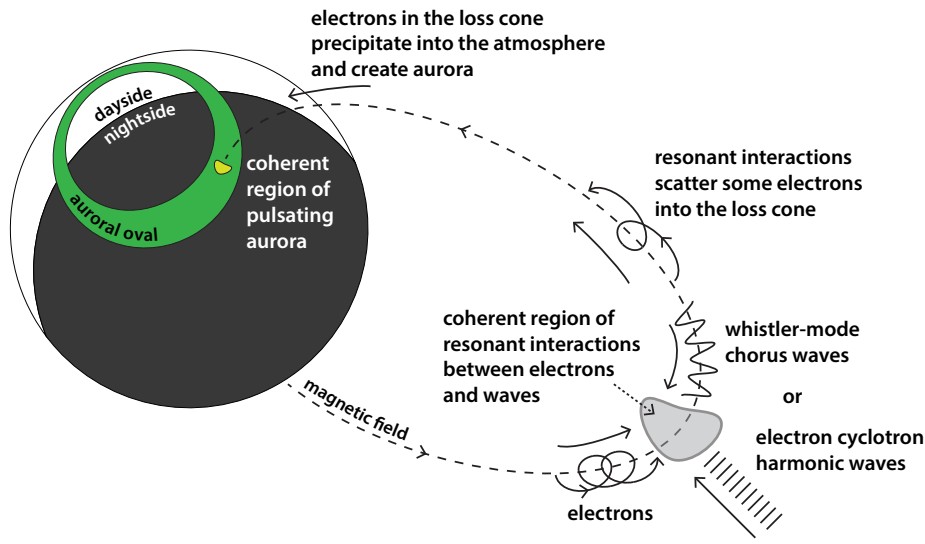

**Figure 1.** Lower-band chorus and ECH are able to undergo cyclotron resonance with equatorial electrons. Many of these interactions occur within a coherent region, scattering some electrons into the loss cone. The electrons with pitch angles in the loss cone precipitate into the atmosphere and produce aurora. The shape of the coherent region where the wave-particle interactions occur is reflected in the shape of the corresponding auroral feature. The pulsations are created when a mechanism modulates the growth rate of the plasma waves. The dim "off" state of pulsating aurora corresponds to times when there are fewer resonant interactions occurring, while the opposite is true during bright "on" states.

with plasma waves at that point in space. The opposite is true during on states. Many details of this process, such as the types
of waves involved and the mechanisms and conditions that modulate their growth and decay, are areas of active research.

Two types of plasma wave have been confirmed to resonate with equatorial electrons and produce pulsating aurora: lower-band whistler-mode chorus (Nishimura et al., 2010, 2011) and electrostatic electron cyclotron harmonic (ECH) waves (Fuk-izawa et al., 2018). Whistler waves are short audio-frequency range plasma waves which either continuously increase or decrease in frequency. Whistlers often occur in rapid succession, forming chorus packets which appear primarily in two frequency
bands below the electron gyrofrequency. These bands are called *lower* and *upper*, and they are located at frequencies below and above half of the electron gyrofrequency, respectively. For cyclotron resonance to occur between chorus and equatorial electrons, they must be oppositely directed such that the wave frequency is Doppler-shifted to be a multiple of the electron gyrofrequency. Whistlers are circularly polarized, so their electric field aligns with the helical trajectory of the electrons during resonance. ECH waves are electrostatic Bernstein mode waves which occur in bands between multiples of the electron gyrofre-
quency. They are longitudinal waves and propagate nearly perpendicularly to the magnetic field. Resonance between ECH and electrons is analogous to adding pushes in and out of phase with the rotation of a spinning bicycle tire to modify its motion. Figure 1 illustrates the creation of pulsating aurora via chorus and ECH.





Complimentary studies by Li et al. (2011b, a) identified a few mechanisms through which chorus growth and decay could be modulated. Li et al. (2011b) found that ultra low frequency (ULF) waves can affect chorus growth by modulating the relative

number of resonant electrons or the electron anisotropy. Li et al. (2011a) found that density enhancements and depletions correlate with chorus intensity and are capable of modulating chorus growth.

The difficulty of associating in situ plasma observations with specific auroral features varies depending on the distance of the spacecraft from Earth. In the ionosphere, magnetic field models are accurate enough for this task, but at higher altitudes the magnetic field becomes increasingly complex and variable. In the nightside equatorial plane where pulsating aurora ob-

servations map to, approximately between 4 and 15 $R_E$ (Grono and Donovan, 2019b), magnetic field models are reasonably accurate but are no longer sufficient for associating in situ measurements with specific auroral features. Enough uncertainty can be produced by varying magnetic field models and input parameters that the magnetic footpoint can be moved between auroral features.

Nishimura et al. (2010, 2011) demonstrated a technique capable of definitively identifying the drivers of specific pulsating

auroral features, and therefore the true footpoint of the observing satellite. Their novel strategy was to recognize that if the bright and dim states of pulsating aurora corresponded to plasma wave modulations, then integrated plasma wave power should correlate with auroral brightness. Reassessment of their events indicates that they are a combination of APA and PPA auroras.

This technique requires magnetic field power spectra measurements of plasma waves and coincident auroral images. In the case of Nishimura et al. (2010, 2011), the power spectra were integrated over the frequency range of lower-band chorus, which

is often generalized as between 5 and 50 % of the electron gyrofrequency. The resulting time series of chorus power must be aligned to the auroral images, within which each pixel forms a time series of auroral brightness. The time series of each of pixel is then cross-correlated with chorus power to produce a correlation value for each pixel. Pixels with high correlations are then mapped into latitude and longitude. When these highly correlated pixels form coherent regions in the ionosphere, they identify the specific auroral feature being produced by the observed plasma waves. This coherent correlated region thus corresponds to

the true magnetic footpoint of the spacecraft.

By observing a nearly one-to-one correspondence between auroral luminosity and integrated wave power, Nishimura et al. (2010, 2011) confirmed that lower-band whistler-mode chorus waves can resonate with electrons with kinetic energies on the order of 100 eV to 10 keV, and are in many cases the primary cause of pulsating aurora. So far, this method has been used to confirm that lower-band whistler-mode chorus and ECH waves (Fukizawa et al., 2018) can cause pulsating aurora.

The aurora is a key tool for remote sensing the large-scale, dynamic plasma processes that the magnetosphere is host to. Pulsating aurora, despite being widespread, is often discussed as if it just one phenomenon and this has restricted the ability to use it to study the magnetosphere. The specific conditions and processes that dictate the formation of different pulsating auroras are unclear, but in situ measurements of their source regions would provide essential information for understanding their differentiation. A significant motivation of this study is that cross-correlating auroral brightness and wave power does

not regularly produce a coherent or reliable magnetic footpoint for space-borne measurements. Certain difficulties with this technique will be outlined here. Between this and the uncertainty in magnetic field mappings, associating in situ observations with specific auroral features is highly challenging. Furthermore, conjunctions between high-Earth orbit spacecraft and images





of specific types of aurora are fundamentally rare. This study suggests a method of avoiding these issues by analyzing the types of chorus structuring one would expect to observe when a spacecraft is conjoined with each type of pulsating aurora. The

results of this analysis are used to constrain the region of the magnetosphere where the processes and conditions responsible for the structuring of PA and PPA can exist.

## 2   Reliability of cross-correlating chorus power with auroral brightness

The original goal of this study was to find conjoined optical and in situ observations of pulsating auroras and compare the conditions and processes observed in their magnetospheric source region. These plasma measurements would then be used

to identify causes of pulsating aurora differentiation. Conjunctions were identified between three Time History of Events and Macroscale Interactions during Substorms (THEMIS) spacecraft, those designated THEMIS A, D, and E (Sibeck and Angelopoulos, 2008), and the all-sky imagers (ASI) comprising the ground-based component of that mission (Donovan et al., 2006; Mende et al., 2008).

The THEMIS spacecraft have elliptical, near-equatorial orbits that precess around Earth, making them well-situated to pro-

vide observations of plasma parameters within the source regions of pulsating auroras. Their Digital Fields Board (DFB; Cully et al., 2008) instruments provide plasma wave power spectra by digitizing and processing waveforms output by the Electric Field Instrument (EFI; Bonnell et al., 2008) and Search Coil Magnetometer (SCM; Roux et al., 2008) instruments. The frequency and temporal resolution of the spectral data products are dependant on the operational mode of the instrument, and for this study we utilized the fast survey operational mode whose cadence varies from 1 to 8 seconds. Fluxgate Magnetome-

ter (FGM; Auster et al., 2008) observations were used to calculate electron gyrofrequencies. This instrument measures the background magnetic field with a resolution of 0.01 nT and can detect low-frequency fluctuations up to 64 Hz.

The THEMIS ASIs image the aurora in "white light" at a 3 second cadence on a $256 \times 256$ pixel CCD and have been operating continually for over 10 years to amass tens of millions of images. Following observations of Jones et al. (2011), it can be surmised that pulsating aurora has been observed within on the order 10 % of these images (Grono et al., 2017).

We identified 30 conjunctions between THEMIS satellites and ASI which were coincident with pulsating aurora. These conjunctions were spread across 23 days and 8 imagers, meaning that some days saw multiple spacecraft pass through a single imager. Most of these conjunctions were quite long and could be separated into many subevents. The divisions between subevents were generally chosen to occur at wider gaps between chorus packets, as these provided a natural separation point in the THEMIS magnetic field power spectra. Ultimately, we found 102 subevents whose durations were typically between 5 and

10 minutes but could be as short as 2 minutes and as long as $\sim$30 minutes if no natural separation point was apparent. Within each subevent, the highest 2 minute cross-correlation between auroral brightness and chorus power was determined. This was done by iterating over each possible 2 minute window for each pixel within the subevent, although the black pixels bordering the cameras' fields of view were ignored. The complete list of subevents is available in Grono (2019)

It is difficult to define exact criteria for what the output looks like when this technique "works". In the best cases there is only

one coherent region, like Figure 2a, or one region that was obviously more strongly correlated than the others. In other cases,





there could be no coherent correlated region at all, such as in Figure 2d. Further, often there were multiple regions correlating approximately as well as each other, an example of which is Figure 2g.

For our purposes, we trusted the output of this technique when any highly correlated, coherent regions were relatively near each other. That is, we did not necessitate that there be only one high correlated region, but that if there were multiple, they

all indicated the spacecraft was in approximately the same position and not separated by a large fraction of the ASI field of view. Furthermore, the locations of these regions had to be robust to small changes to the time period being correlated. Only 16 subevents met these criteria.

All subevents featured lower-band chorus, and a handful had upper-band chorus in addition. There does not appear to be a connection between the type of pulsating aurora and the type of plasma wave.

We were not able to establish which conditions are necessary for this technique to work effectively, but we do know they can change rapidly. In Figure 2, events 1 and 2 are separated by approximately 15 minutes and the output of the cross-correlation switches from producing a single highly correlated, coherent region to there being no highly correlated coherent region at all. There are ASI and chorus observations between these two subevents, but the ASI is partly obstructed by generator exhaust during this period.

This type of variability can also take the form of single, highly correlated, coherent peaks moving to entirely different auroral features on similar time scales. By other standards one may be able to judge more subevents as successful, but we found that it could not be relied upon to compare the source regions of pulsating auroras. Ultimately, it is not currently possible to reliably associate an auroral feature with observations from high Earth-orbit spacecraft. However, if integrated chorus power can have a nearly one-to-one correspondence with auroral intensity (Nishimura et al., 2010), then auroral intensity should be able to act

as a proxy for chorus power.

## 3 Inferring proxy in situ chorus time series from ASI image sequences

Figure 3 outlines the process through which auroral brightness can be used as a proxy for chorus power. This figure contains a set of subfigures for each type of pulsating aurora which consist of three panels: a *pseudo-conjunction*, its complete orbit, and its along-track auroral intensity. The leftmost panel, the pseudo-conjunction, is a fake 30 minute section of a THEMIS orbit

deliberately constructed to pass through specific auroral features.

The procedure for calculating the orbits which create these pseudo-conjunctions begins by randomly selecting 30 minutes of ASI data during a pulsating aurora event and then randomly selecting an image from that period. This image is then mapped out into latitude and longitude and a point is randomly selected within it. We constrained the location of this point to pixels above 30 degrees elevation in order to avoid most undesirable features near the edge of certain ASIs' fields of view. Moreover,

this point must be restricted to latitudes that could map to a location within the apogee of THEMIS. Any farther out and the satellite would not be able to reach it.

To do this, an orbit approximately matching those of the THEMIS spacecraft, having an apogee of 12 $R_E$ and a perigee of 1.15 $R_E$, is calculated following Kepler's laws of planetary motion. For the sake of simplicity, the orbit is constrained to the





# Identifying magnetospheric drivers of pulsating aurora by cross-correlating wave intensity and brightness is unreliable

**Event 1**    *Cross-correlation produces single coherent area*
*Best 2 minute correlation between 03:46:00 - 03:51:00*

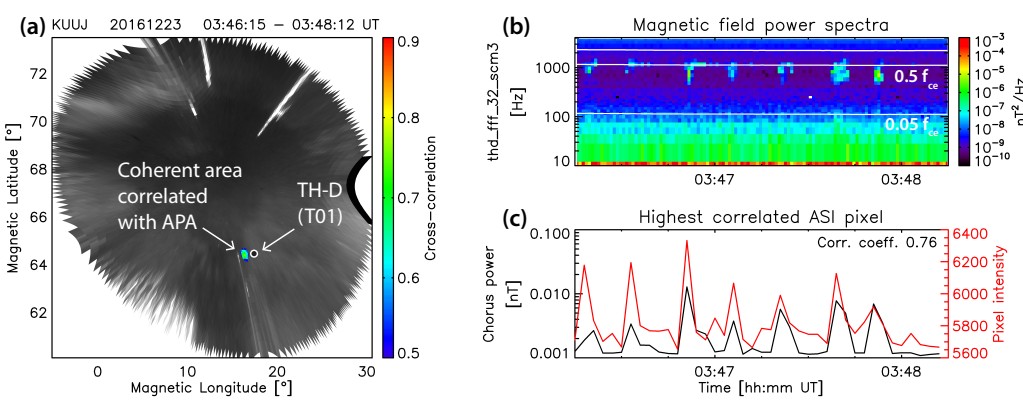

**Event 2**    *15 minutes after Event 1... cross-correlation no longer produces coherent area*
*Best 2 minute correlation between 04:02:30 - 04:06:00*

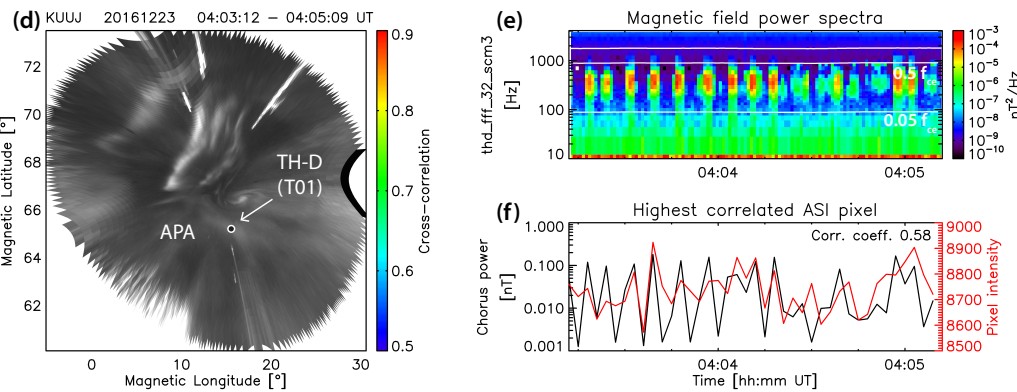

**Event 3**    *Multiple patches have coherent cross-correlation areas*

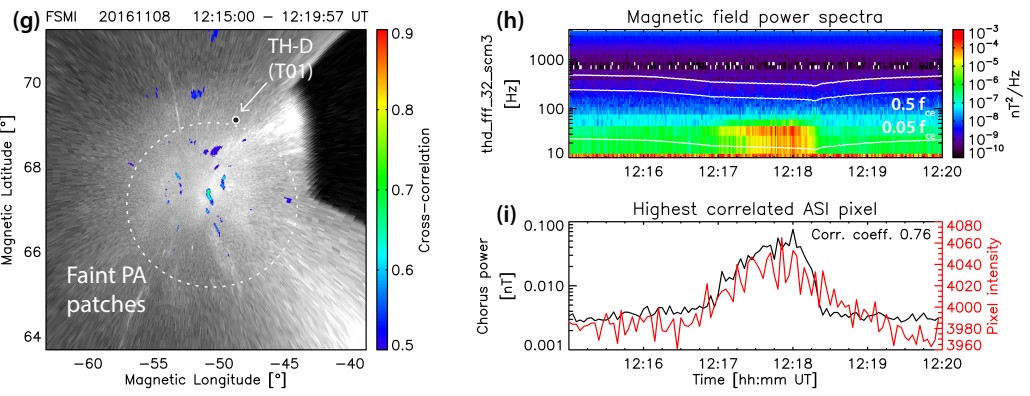



**Figure 2.** Cross-correlating wave power with auroral brightness is a powerful tool when it works, but it is unreliable. Separated by only 15 minutes, a single coherent correlated area exists during event 1 but not in event 2. In event 2, the highest correlated pixel is located below 5 degrees elevation, the threshold below which pixels were not drawn, and is located near ∼25 degrees MLON and ∼58 degrees MLAT. Event 3 cross-correlates a ∼1-minute-long chorus packet with a sequence of ASI images of PA patches. Such chorus packets do not have a sufficiently unique signature in their integrated power time series to create a single coherent correlated region. Chorus power was integrated between 5 and 50 % of the electron gyrofrequency ($f_{ce}$). Only pixels with correlations stronger than 0.5 are coloured. The white and black markers indicating the footpoint of the THEMIS D spacecraft is calculated using the T01 model (Tsyganenko, 2002).

XY GSM plane. The location of the central pixel of the ASI field of view is mapped to the equatorial plane with the T89 model
(Tsyganenko, 1989) assuming an altitude of 110 km, passing values for the planetary K-index (Kp) and solar wind velocity X GSE component. These parameters were averaged over the 30 minute period in order to limit the possibility of bad data. The THEMIS-like orbit is then rotated such that its major semi-axis is in line with the XY GSM location of the mapped central pixel. The apogee of this orbit is then mapped back to 110 km altitude, and this latitude acts as the poleward boundary of the region where a point is randomly selected.

Once a point is selected, it is then mapped out to the equatorial plane using the T89 (Tsyganenko, 1989) model, again assuming an initial altitude of 110 km. Another THEMIS-like orbit is then rotated to pass through the location traced from the all-sky image. Whether the orbit is rotated to pass through the mapped point while the spacecraft would be travelling toward or away from the Earth is decided randomly. Time steps between each point in the orbit can be calculated following Kepler's laws, which combined with the exposure time of the initial image can be used to determine the satellite location during each
exposure in the 30-minute period. These locations are then traced back to the initial 110 km altitude assumed for these all-sky images to determine the spacecraft's footpoint in each image.

The leftmost panels of Figure 3 show the initial randomly chosen image with the randomly chosen point marked as a red dot. The blue lines represent the path of the satellite footpoint during the 30 minutes of ASI data comprising the pseudo-conjunction. The middle panels show the complete fake orbits in XY GSM coordinates. The blue segments indicate the portion of the orbit
that aligns with the 30 minute period of optical data. However, these segments are quite small and are partly obstructed by the red dots which mark location that the initial randomly selected point was traced to. The rightmost panels are the along-track auroral intensity of the pseudo-conjunction. In each image, the nearest pixel to the fake satellite location is identified and the intensities of these pixels are arranged in time series as the along-track brightness. The red dot indicates the intensity at the satellite footpoint in the initial randomly chosen images.

**4 Proxy in situ chorus time series for different types of pulsating aurora**

Since Nishimura et al. (2010) discovered that there can be a nearly one-to-one correspondence between chorus power and auroral intensity, then auroral intensity should act as a proxy for chorus power. Pseudo-conjunctions could be used to determine whether APA, PPA, or PA have any unique characteristic features in their associated plasma waves. If they do, then this





# Inferring chorus power from pulsating aurora



**Figure 3.** If there can be a one-to-one correspondence between chorus power and auroral brightness (Nishimura et al., 2010, 2011), along-track brightness can be used to infer chorus power. This is a comparison of what chorus power could look like for amorphous pulsating aurora, patchy pulsating aurora, and patchy aurora. The red dot indicates the location and time that was used to constrain the constructed THEMIS orbit.





information could be used to deduce what type of pulsating aurora that chorus observations may correspond to without the need for coincident optical observations. Since it proved so difficult to find definite conjunctions between THEMIS satellites and ASI for each type of pulsating aurora, pseudo-conjunctions could be useful tool for circumventing this problem.

Figure 4 shows 18 examples of along-track auroral brightness for pseudo-conjunctions with APA, PPA, and PA, 6 examples for each type. These examples are unique from those of Figure 3, making for a total of 7 each and 21 shown in this paper altogether. They are a selection of a dataset which was created by randomly generating 30 conjunctions for each of 29 1-hour pulsating aurora events spread across 19 days and 7 imagers. The conjunctions included here were chosen to avoid showing multiple of the same type from the same date and site. Each event had one or two dominant types for which they were best suited to generate a conjunction. The necessary information for recreating these pseudo-conjunctions is available in Grono (2019).

## 5 Using chorus packets to locate the PA source region

If pulsating aurora brightness is a proxy for chorus power, then PA patches, which mostly remain in their bright state, should be caused by the absence of chorus modulation within their source regions. PA should thus be associated with unmodulated chorus packets persisting for the length of time it takes for a satellite to traverse the source region of a patch. Based on Figure 3c and the PA panels of Figure 4, this length of time is on the order of minutes, with most transits taking being approximately between 1 to 5 minutes. By identifying such chorus packets in THEMIS measurements, it should be possible to identify the region where the structuring processes and conditions that are responsible for PA and PPA are able to occur.

In general, a chorus packet is a burst of rising or falling tone whistler waves. However, for the purpose of algorithmically identifying them in THEMIS magnetic field power spectra, we define them as a structure in the spectra located between 5 and 80 % of the electron gyrofrequency whose integrated power is greater than 20 pT. We chose to integrate over the frequency ranges of lower and upper-band chorus because, while only lower-band chorus and ECH have been confirmed to produce pulsating aurora, upper-band chorus should be able to as well (Ni et al., 2008). The lifetime of a chorus packet is defined here as the time the intensity first becomes greater than 20 pT subtracted from the time it passes back below this threshold. This definition is illustrated in Figure 5.

Chorus packets were searched for in the 1 second cadence fast survey data by integrating the magnetic field power spectra between 5 and 80 % of the electron gyrofrequency and identifying when it crossed over the 20 pT threshold. Between 2014 and 2016 inclusively, 1 875 302 packets were found over 769, 819, and 837 days of THEMIS A, D, and E data, respectively. We were primarily interested in nightside activity but wanted to be able to estimate for how long PA and PPA structuring can persist after sunrise. To this end, we searched for events between 22 and 14 magnetic local time (MLT), avoiding the dusk-sector due to the presence hiss related to enhanced density plumes (Meredith et al., 2004). Measurements inside 3 $R_E$ were ignored for being too equatorward of the auroral zone and unrelated to pulsating aurora. In addition, events near the dayside magnetopause were excluded due to intermittent fluctuations of the magnetopause standoff distance which cause the spacecraft to cross into the magnetosheath.





**Figure 4.** Samples of pseudo-conjunction along-track auroral brightness for APA, PPA, and PA. Marker colour serves no other purpose than to help visually distinguish events of each type.





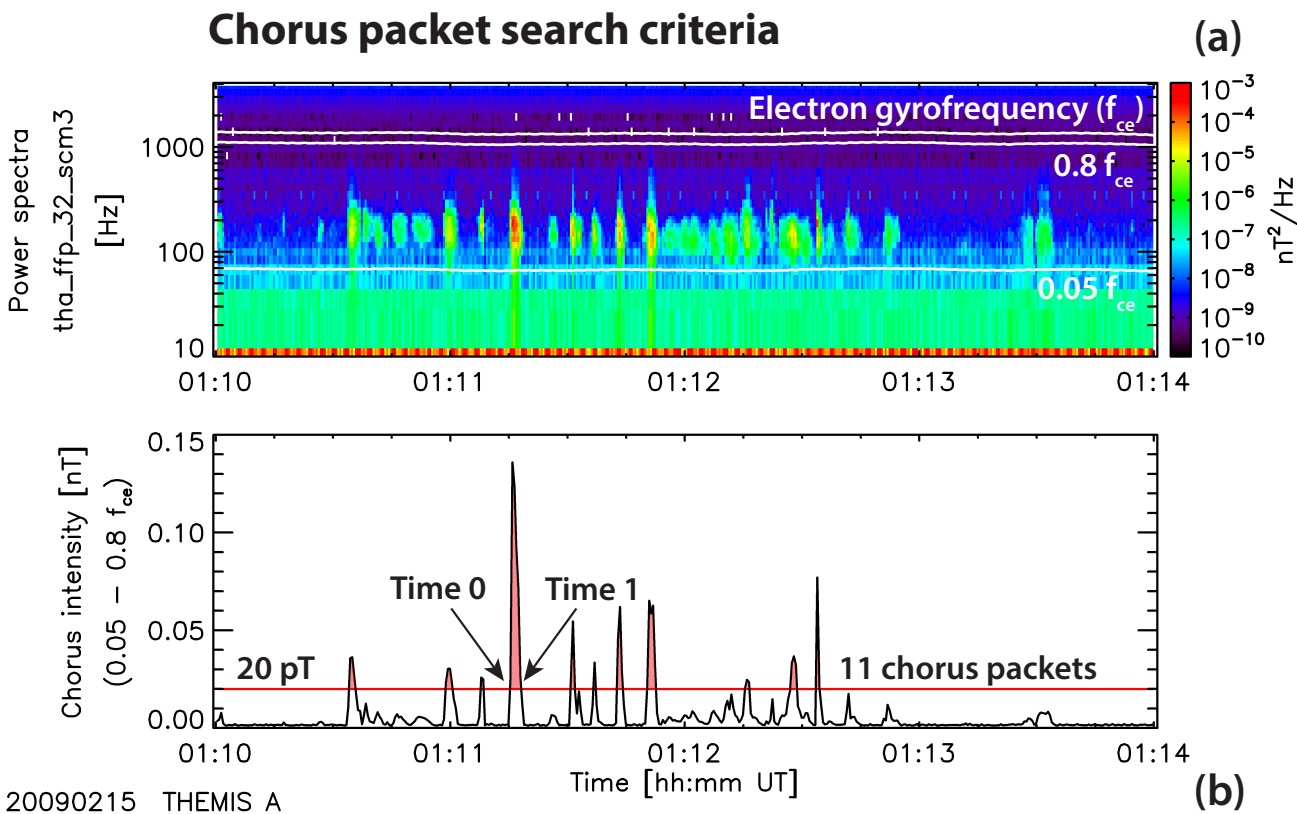

**Figure 5.** Chorus packets, as they were searched for in magnetic field power spectra such as that in panel (a), are individual bursts of wave power that are greater than 20 pT. The lifetime of a chorus packet is the time the wave intensity, integrated between 0.05 and 0.8 $f_{ce}$, becomes greater than 20 pT (time 0) subtracted from the time it drops back below 20 pT (time 1). The 11 chorus packets that surpassed this threshold in panel (b) are highlighted in red.

Figure 6 shows the occurrence rate distributions of two types of chorus packets. Panel 6a depicts the distribution of chorus packets with lifetimes greater than 0 seconds and less than or equal to 1 minute, while panel 6b shows the distribution of "longer-lived" packets having lifetimes greater than 1 minute and less than or equal to 5 minutes. Packet occurrence rate is
calculated by dividing the number of chorus packets observed in a bin by its total amount of THEMIS FFT instrument "on-time" in minutes. The instrument on-time is the length of time it was operational for while the spacecraft was located within a bin. Due to the 1 second cadence of the fast survey operational mode of the instrument, the shortest possible packet lifetime is 1 second.

Panel 6a shows that the occurrence rate of short-lived chorus packets generally increases with increasing MLT. Between 22
to 0 MLT, occurrence is relatively low between 3 and 10 $R_E$ and then increases suddenly beyond that point until the end of data collection near 12 $R_E$, the outer limit of THEMIS orbits. Between 0 and 2 MLT, the situation is mostly the same, but occurrence drops to zero, or nearly zero, between 9 and 11 $R_E$. Afterward, this depletion disappears and from 2 until 4 MLT occurrence





# Inferring pulsating aurora source regions from chorus packet lifetime

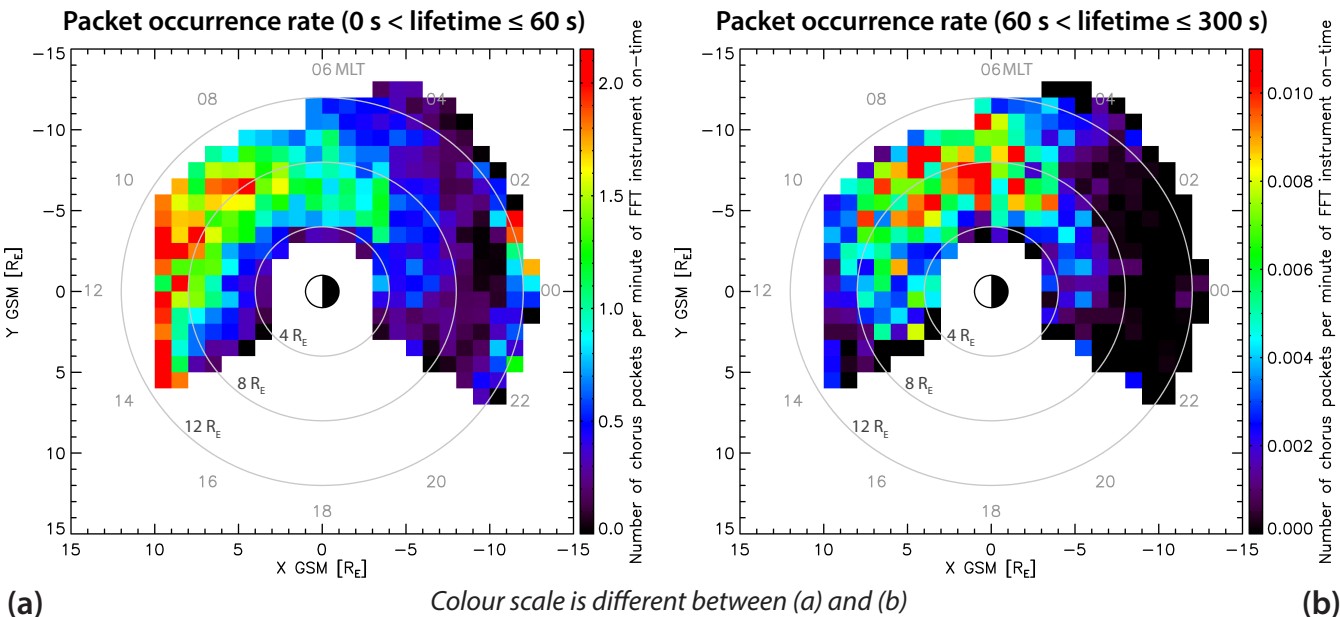

**Figure 6.** If chorus modulation is responsible for auroral pulsations, then patchy aurora should correspond to long-lived chorus packets and not short ones. Thus, the distribution in (a) of chorus packets with lifetimes between 0 and 60 s should exclude most patchy aurora and represent the source region of amorphous pulsating aurora, and possibly include patchy pulsating aurora. The distribution in (b) of packets between 60 and 300 s in lifetime corresponds to the source region of patchy aurora, but again possibly includes patchy pulsating aurora. Occurrence rate is calculated by dividing the number of chorus packets in a bin by its total FFT instrument on-time. White indicates no data coverage.

drops off beyond 12 $R_E$. At 4 MLT, occurrence increases suddenly everywhere beyond 4 $R_E$, especially between 4 and 8 $R_E$. This persists until 8 MLT when occurrence increases significantly past 12 $R_E$. The short-lived packets occur most frequently

on the dayside from 9 to 14 MLT between 8 and 10 $R_E$. Within this time period, occurrence increases with increasing radius.

The longer-lived packets shown in panel 6b are constrained to a semi-annular region which is narrowest on the nightside from 22 to 3 MLT when it is primarily between 4 and 8 $R_E$. During this time, the occurrence rate of the longer-lived chorus packets falls nearly to zero beyond 8 $R_E$. Between 3 and 5 MLT, the occurrence region widens and is seen to drop off slightly before 12 $R_E$ where data collection stops. After 5 MLT, data from 10 to 12 $R_E$ begins to be removed from the distribution to

avoid regions where spacecraft occasionally cross into the magnetosheath. The semi-annular region continues up to the limit of data collection from 6 to at least 8 MLT, although possibly until 10 MLT. At this point, occurrence drops rapidly beyond 8 $R_E$. Slightly before 14 MLT, longer-lived chorus packets promptly decrease to zero at most distances data was collected at. The occurrence of longer-lived chorus packets peaks approximately between 6 and 10 MLT.





## 6  Discussion

Figures 3 and 4 allow us to make certain inferences about how the plasma waves, likely lower-band chorus, could be structured for each type of pulsating aurora. The along-track intensities of the APA pseudo-conjunctions typically exhibit significantly more scatter than those of PPA and PA. Varying degrees of coherent structuring indicative of auroral features consistently brighter than the background can be seen in the APA brightness time series. The PA along-track intensities have the least scatter, and distinct auroral features are always clearly visible. The characteristics of PPA intensities fall in between these two

extremes, featuring a mixture of coherent structuring and high scatter.

Summarizing these results in the context of plasma wave power, pulsating auroras may be associated with chorus structuring resembling the following descriptions. APA could correspond to observations of long periods of sequential short-lived chorus packets. PPA would also generally be associated with many sequential short-lived chorus packets, and these packets may be arranged in bursts that persist for the length of time it takes a spacecraft to traverse the PPA source region. PA could correspond

to observations of chorus packets unmodulated for the duration of a spacecraft's transit through the PA source region.

Of course, reality is too complicated for this to be a reliable heuristic. Different pulsating auroras often coexist and as a spacecraft transits from one type to another, the chorus observations would become complicated enough that it would be challenging to identify any unique signature. Due to the variety of appearances APA along-track intensities can have, it may not be possible to eliminate the chance that chorus observations are of APA. However, we can draw one key conclusion from

this analysis: PA should be associated with longer-lived chorus packets that feature little or no modulation.

There are two main issues that could prevent this conclusion from being useful. First, APA is far more common than PA (Grono and Donovan, 2019b) and within the 7 APA pseudo-conjunctions in Figures 3 and 4, certain structures could realistically be confused with PA. Second, our criteria for searching for chorus packets does not exclude the possibility that the packet could be highly modulated while it remains above the 20 pT threshold.

However, PPA and PA exhibit the same structuring (Grono and Donovan, 2018) and have similar distributions both in the ionosphere (Grono and Donovan, 2019b) as well as relative to the proton aurora (Grono and Donovan, 2019a). This suggests that the same structuring processes may be responsible for both, and that if you were to infer the source region of PA from a distribution of longer-lived chorus packets, you would also be identifying the source region of PPA. Thus, it is not a problem if our chorus packet criteria also include PPA events. This would have the added benefit of estimating how long past sunrise PA

and PPA can occur. Based on observations from ASI data (Grono and Donovan, 2019b), which cannot continue past dawn, PA occurrence peaks between 4 and 5.5 MLT and PPA between 5 and 6 MLT. Furthermore, it is not uncommon to see pulsating aurora events featuring PPA and PA where these auroras persist until camera shutdown. Despite all of this, if APA is frequently caused by these same longer-lived chorus packets, then their distribution would be useless for identifying the PA and PPA source region since APA can occur in a broader region than PPA and PA (Grono and Donovan, 2019a).

Conjunctions between THEMIS spacecraft and ASI during pulsating aurora are relatively rare, and of these, few can be definitively associated with a specific type. Considering that the probability of occurrence for PPA and PA peaks at 21 and 29 % (Grono and Donovan, 2019b), respectively, conjunctions with these types are particularly hard to find. However, there





is a more fundamental issue intrinsic to PA that makes identifying conjunctions by cross-correlating plasma wave power and auroral brightness essentially impossible.

Pulsating aurora events featuring PA will have many patches. Even if one knew with certainty that a spacecraft traversed the source region of a PA patch and measured the expected long-lived chorus packet, the wave power would not correlate with a unique auroral feature, but many. Observations of singular chorus packets produce many coherent correlated areas. During an event with multiple PA patches, some amount of pixels in each of them tend to correlate equally well, making it difficult to find a unique signature. Event 3 in Figure 2 demonstrates this. A conjunction where multiple PA patches were transited would

be even more difficult to find and would not necessarily make cross-correlating with brightness work better. A window of time long enough for a spacecraft to transit two patches would likely be too long for any single pixel to correlate well with wave power. The farther a spacecraft travels, the less accurately any single pixel could reflect the wave power observed in situ.

For this reason, it is particularly useful to be able to infer when a satellite may be transiting a PA source region. To determine whether chorus packets with lifetimes between 1 to 5 minutes could be the driver of PA patches, we need to compare the

locations of the chorus packets to a mapping of where PA structures are observed. Grono and Donovan (2019b) used the T89 model (Tsyganenko, 1989) to trace their distribution of PA occurrence into the equatorial plane of the magnetosphere.

The mapped locations of patchy aurora as determined by Grono and Donovan (2019b) are shown in Figure 7. All bins found by Grono and Donovan (2019b) to have more than one PA event map to it are filled in red. These bins are plotted together with those of Figure 6b, the distribution of chorus packets with lifetimes between 1 and 5 minutes. The chorus packets bins are

coloured blue if the packet occurrence rate was greater than 0.001 packets per minute. Purple bins represent the region where these two distributions overlap.

On the nightside of the magnetosphere, these two regions have a similar spatial extent. Occurrences terminate at ∼23 MLT and are approximately constrained between 4 and 8 $R_E$ near magnetic midnight before widening to ∼12 $R_E$ near dawn. Discrepancies between the two distributions are in part attributable to inaccuracies in the magnetic field mapping of the PA occurrence

distribution.

While sunrise prevents all-sky cameras from operating past dawn, the chorus packet distribution suggests that the processes which are responsible for the structuring of PA and PPA may persist for much longer, possibly until ∼14 MLT. In fact, Figure 6b suggests that the structuring process could be most prevalent between 6 and 10 MLT. During this time, the occurrence rate of longer-lived chorus packets is greatest. Figure 7 also suggests that APA features associated with longer-lived chorus packets

are less common compared to PA and PPA, at least outside of mapped PA region on the nightside between 23 and 6 MLT. If this was not true and such APA events were common elsewhere, then we would not have expected the occurrence rate of such chorus packets to fall to 0 packets per minute — or nearly zero — outside of PA occurrence region.

Investigating the characteristics of these longer-lived chorus packets would help determine whether multiple processes are contributing to this distribution. Furthermore, chorus packets were only searched for between 22 and 14 MLT to avoid hiss

in enhanced density plumes (Meredith et al., 2004), so the full extent of the semi-annular chorus packet distribution could be unclear. Despite this, occurrence falls substantially at the edges of the covered region to indicate that the limits to its extent are being observed.

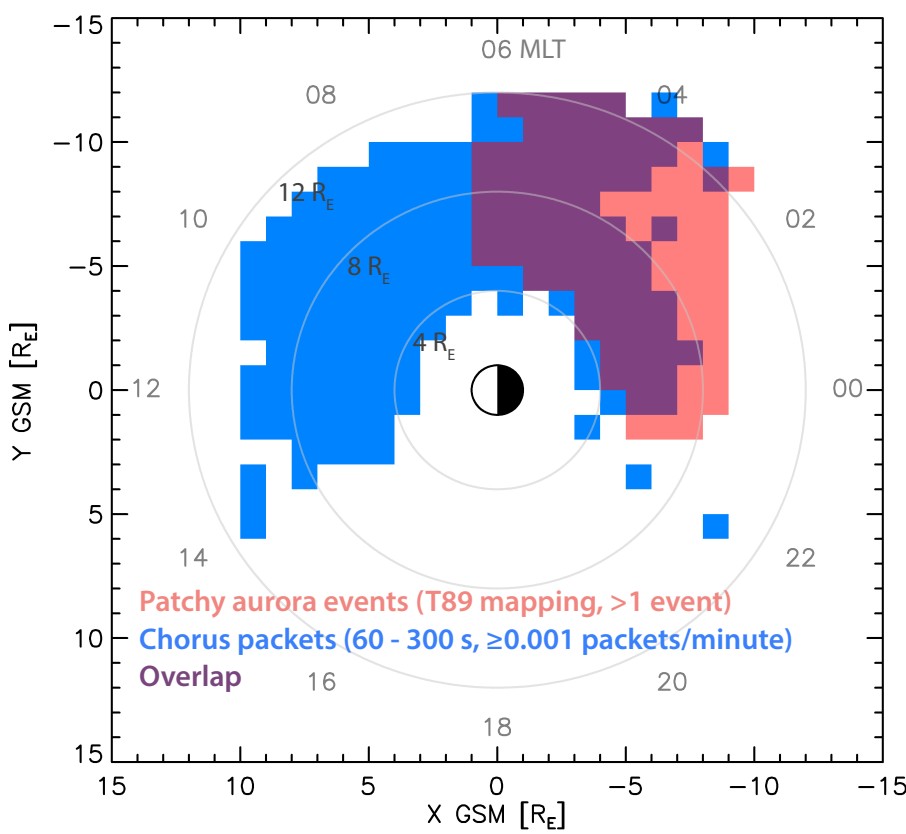

**Figure 7.** Comparison of Figures 6b and the T89 (Tsyganenko, 1989) mapped PA occurrence region shown in Figure 4c of Grono and Donovan (2019b). The bins of Grono and Donovan (2019b) are filled red if they have more than 1 event in them, and the bins of 6b are filled blue if they have more than 0.001 packets per minute. Purple bins indicate where the populations overlap.

## 7 Conclusions

In situ plasma measurements of pulsating aurora source regions will be necessary for determining which mechanisms and
conditions are responsible for the different types of pulsating aurora. Unfortunately, it is not currently possible to reliably associate high Earth-orbit in situ measurements with specific auroral features. The only method of definitively determining which pulsating auroral feature a high-Earth orbit spacecraft is conjoined with is to cross-correlate plasma wave power with auroral brightness. As we have shown, this technique is powerful when it works but is ultimately unreliable. It appears that there are specific conditions that result in a one-to-one correspondence between chorus power and auroral brightness, and
such conditions may be relatively uncommon. While magnetic field models are accurate enough to trace spacecraft to specific





auroral features in the ionosphere and innermost magnetosphere, they are too uncertain in the source regions of pulsating auroras.

To investigate differences in the source regions of pulsating auroras we have proposed an alternative to magnetic field mapping and cross-correlation. If integrated chorus power and auroral brightness can have a nearly one-to-one correspondence
(Nishimura et al., 2010), then auroral intensity can be used as a proxy for the chorus power within a pulsating auroral feature. Fake conjunctions can be calculated to specifically pass through the types of pulsating aurora, and from them along-track auroral intensities can be constructed. Structuring in the along-track brightnesses of these pseudo-conjunctions should reflect that of the plasma waves in the pulsating aurora source region the satellite would be passing through.

PA should thus be driven by mostly unmodulated chorus with packet lifetimes corresponding to the time it takes to transit the
source region of a patch. Based on the along-track intensities of the pseudo THEMIS conjunctions presented in Figures 3 and 4, this was judged to be on the order of 1 to 5 minutes. The locations of chorus packets with this range of lifetimes are compared to the locations of patchy aurora events mapped to the equatorial plane from Grono and Donovan (2019b) in Figure 7, and can be seen to overlap to a large extent.

Figure 7 also shows that chorus packets with these longer-lived lifetimes drop to 0 packets per minute, or nearly so, outside
of the PA occurrence region. This indicates that APA, which is by far the most common pulsating aurora (Grono and Donovan, 2019b), does not appear to be often caused by longer-lived chorus packets, at least on the nightside.

PA and PPA have been shown Grono and Donovan (2019a) to be predominantly constrained equatorward of the optical b2i (Donovan et al., 2003), the ionospheric marker of the isotropy boundary (Sergeev et al., 1983). This boundary is the transition between stably trapped protons and those affected by stochastic scattering. Our result concurs with the observations of Grono
and Donovan (2019a), providing further evidence that the processes and conditions responsible for the characteristic structuring of PA and PPA are constrained to the inner magnetosphere.

The aurora is one of our main tools for remote sensing the large-scale dynamics of our vast magnetosphere. Auroras are signatures of specific mechanisms and conditions that exist in the magnetosphere or are born out of magnetosphere-ionosphere coupling. Pulsating auroras are regular morning-sector auroral phenomena, and in order to use them to study magnetospheric
plasma processes, it is necessary to understand the nature of their differences. Ultimately, in situ observations of the plasma conditions in the source regions of pulsating auroras will be necessary for discovering the causes of their differentiation. This study is a step toward being able to circumvent the inherent difficulties of associating high Earth-orbit in situ observations with specific auroral structures which could allow the structuring processes involved in PA and PPA to be studied on a large scale.

*Data availability.* The information necessary for reproducing the complete set of pseudo-conjunctions discussed in this paper as well as
the list of THEMIS spacecraft and ASI conjunctions are available in Grono (2019). THEMIS ASI data is available from http://data.phys. ucalgary.ca/sort_by_project/THEMIS/asi/stream0/. THEMIS spacecraft data was accessed and processed using SPEDAS, see Angelopoulos et al. (2019). Conjunctions between THEMIS spacecraft and ASI were identified using Swarm-Aurora, available at https://swarm-aurora. com/. Planetary K-index data was retrieved from the National Oceanic and Atmospheric Administration Space Weather Prediction Center at





ftp://ftp.swpc.noaa.gov/pub/indices/old_indices/. Tsyganenko model parameters obtained via Operating Missions as Nodes on the Internet
(OMNI).

*Author contributions.* EG built the dataset, analyzed it, designed the figures, and wrote the text. ED is his supervisor and assisted with the analysis and editing.

*Competing interests.* No competing interests are present.

*Acknowledgements.* This research was supported by grants from the Natural Science and Engineering Research Council (NSERC) of Canada
and Danish Technical University (DTU). Thanks to Emma Spanswick, Harald Frey, and Stephen Mende for All-Sky data from the NASA
Time History of Events and Macroscale Interactions during Substorms (THEMIS) mission.



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
