# Peer review of "Inferring the source regions of pulsating auroras"

_Annales Geophysicae, 2019_

## Referee Comment (RC1) · Anonymous Referee #1 · 24 Jan 2020

This manuscript demonstrates a new approach to infer the characteristic temporal properties of in situ chorus waves associated with pulsating aurora based on the along-track auroral brightness of pseudo THEMIS conjunctions, and then attempts to locate the magnetospheric source regions of the different types (APA, PPA and PA) of pulsating aurora by identifying the characteristic chorus packets in THEMIS measurements. The results suggest that the occurrence distributions of the short-lived (< 1 min) and longer-lived (1-5 min) chorus packets are associated with the APA/PPA and PA/PPA occurrence regions, respectively, at least on the nightside. Whereas this study has provided some valuable results for understanding the source and generation mechanism of pulsating aurora, for improvements I have some concerns and suggestions, which should be addressed before publication.

Major comments:

(1) I'm a little confused about the definition of "patchy aurora (PA)", which is one of the three types of pulsating auroral patches categorized by Grono and Donovan (2018). Grono and Donovan (2018) defined "PA" as a pulsating aurora which is characterized by stable structures whose pulsations are limited to small regions. The original definition means that PA has more or less a luminosity pulsation. However, the definition seems to contradict what the authors say at Lines 50-51 of this manuscript, "PA is regard as non-pulsating pulsating aurora". Could you explain more explicitly whether PA is pulsating or non-pulsating aurora? In my opinion, the term "non-pulsating pulsating aurora", which confuses readers, should be avoided. Also, if PA is just a diffuse, patchy aurora with no luminosity pulsation, I think that PA should not be regarded as a type of pulsating aurora.

(2) As many early studies reported, chorus waves are often modulated by ULF waves. In this manuscript, however, the authors have not discussed their results in terms of ULF modulation. Using the THEMIS data, is it possible to examine to what extent short-lived (< 1 min) and longer-lived (1-5 min) chorus packets are associated with the modulation by Pc3-Pc5 ULF waves? Otherwise, the authors should discuss the occurrence patterns of both chorus packets in terms of ULF modulation.

Minor comments:

(3) Lines 11-28: The first three paragraphs may not be necessary because the information is too general. So, the authors can remove them and start "Pulsating aurora is ...", "Chorus waves are ...", or something like that.

(4) Line 40: Pulsating aurora occurs not only on the nightside but also on the dayside. Although studies of dayside pulsating aurora are very limited, it would be better to refer to some early studies.

Brekke, A., and H. Pettersen (1971), Some observations of pulsating aurora at

Spitzbergen, Planet. Space Sci., 19, 536–540, doi:10.1016/0032-0633(71)90171-1.

Craven, M., and G. B. Burns (1990), High latitude pulsating aurorae, Geophys. Res. Lett., 17(9), 1251–1254, doi:10.1029/GL017i009p01251.

Wu, Q., and T. J. Rosenberg (1992), High latitude pulsating aurorae revisited, Geophys. Res. Lett., 19, 69–72, doi:10.1029/91GL02781.

Vorobjev, V. G., O. I. Yagodkina, D. G. Sibeck, and P. Newell (1999), Day-time high‐latitude auroral pulsations: Some morphological features and the region of the magnetospheric source, J. Geophys. Res., 104(A5), 10,135–10,144, doi:10.1029/1998JA900158.

(5) Line 180: The authors can delete "(Tsyganenko, 1989)" because it is already sited at Line 175.

(6) Line 218: How did you choose the lower limit (20 pT)?

(7) Lines 244-245: It would be better to discuss comparison of the occurrence distribution with dayside pulsating aurora stated above.

(8) Line 262-264: The authors suggest that APA and PPA could be associated with sequential short-lived chorus packets. However, the occurrence distribution of the short-lived chorus packets, being higher on the dayside than nightside, seems to be somewhat surprising result, because the distribution conflicts with the general pattern of pulsating aurora which occurs preferentially in the postmidnight to early-morning sector. Could the authors explain about that? Moreover, the occurrence rate tends to be higher in the inner magnetosphere (< 8 RE) on postmidnight-morning side, while it is higher in the outer magnetosphere (> 8 RE) on dayside. Do the authors have any interpretation regarding the MLT and L dependences of the occurrence rate?

(9) Lines 308-310: Could the authors see any difference in the mapping result between different magnetic field models (T89, T96, T01, TS07, etc.)?

[Figure]

(10) Could the authors mention how the occurrence distributions of chorus packets varies depending on the intensity of substorm activity?

(11) The authors may also be interested in other recent papers on pulsating aurora:

Kawamura, S., Hosokawa, K., Kurita, S., Oyama, S., Miyoshi, Y., Kasahara, Y., et al. (2019). Tracking the region of high correlation between pulsating aurora and chorus: Simultaneous observations with Arase satellite and ground‐based all‐sky imager in Russia. Journal of Geophysical Research: Space Physics, 124, 2769– 2778. https://doi.org/10.1029/2019JA026496

Nishimura, Y., et al. (2020), Diffuse and Pulsating Aurora, Space Science Reviews, 10.1007/s11214-019-0629-3, 216, 1.

---

## Referee Comment (RC2) · Anonymous Referee #2 · 28 Jan 2020

Summary

Grono and Donovan inferred the source region of pulsating aurora using THEMIS all sky imager and satellite data. The correlation analysis shows that high correlations are not common but when a correlation is found the technique is powerful for finding the source region. Pseudo-conjunctions are used to find the duration of different types of pulsating aurora in the satellite data. The authors conclude that PA is driven by mostly unmodulated chorus but APA is not.

The title and the contents of the paper aren't consistent. The authors misunderstand that PA is pulsating aurora. PA does not pulsate and the spatial distributions presented here aren't appropriate for inferring the pulsating aurora source region. The conclusion that PA is driven by unmodulated chorus is trivial and not new. Furthermore, the paper

has a number of serious assumptions in the analysis method and misinterpretations of the results. The low correlation is likely suffered from technical limitations that the authors used. The paper didn't exclude hiss that occurs on the dawnside and dayside. Additionally, the paper consists of topics that are disconnected from each other. Currently it appears that the paper combines a few different research topics and is therefore unfocused. Considering these major issues, I cannot recommend publication of this paper at this stage.

Major comments

The title states that the main topic of this paper is pulsating aurora but a large portion of this paper deals with aurora that doesn't pulsate. The pulsating aurora definition in this paper (line 49-52) includes PA but the same paragraph says that PA does not pulsate. Figure 6 claims that it shows source regions of pulsating aurora but long-lived chorus doesn't modulate precipitating electrons and therefore isn't related to pulsating aurora. Figure 7 shows the rough connection between the waves with long lifetime and PA, but this connection does not suggest anything about pulsating aurora, because PA is not pulsating aurora. To make the story of the paper consistent, the paper should remove PA or remove "pulsating aurora" throughout the text.

Line 270 The key conclusion on PA is not new. Pulsating aurora (PPA or APA) is known to be caused by rapid wave intensity modulation as shown by the references in the introduction; it's obvious that PA is caused by waves without rapid intensity modulation.

Line 143 The correlation analysis has critical assumptions and technical limitations that aren't considered in the paper. First, the authors require that high correlation areas should be near each other. When more than one high correlation areas are found, one of them can be the high correlations that the authors want to find and the others can occur wherever intensity happens to change similarly. Just there is no good way to find which one the true high correlation is. This situation should be differentiated from the "low correlation" category where a high correlation region doesn't exist.

Line 150-154 The authors state that there is no highly correlated coherent region at all in event 2 because there is no high correlation region even though a high correlation pixel does exist as shown in Figure 2(f). The authors should notice particle injection and dipolarization between event 1 and event 2. The magnetic field geometry changed and therefore the footprint moved between the events. The T01 magnetic field model doesn't consider substorms and cannot express the motion of high correlation region by dipolarization. The apparent disappearance of the high correlation region could simply be because the high correlation region moved away from the model footprint. The author's technique has a limitation that the high correlation region is falsely labeled as uncorrelated when the actual high correlation region moved to low elevation of outside the imager coverage. Events with substorms or solar wind condition changes could move the actual satellite footprint and should be categorized separately. See the second panel at 04:05 UT http://themis.ssl.berkeley.edu/summary.php?year=2016&month=12&day=23&hour=0406&sumType=thd&type=moms

APA involves pulsations faster than 3 s (line 47). The overly smoothed data from the THEMIS imagers does not catch the actual time series of fast pulsation. The low sampling rate could be one of the reasons for low correlation and this instrumental effect is not considered in this paper. APA should be removed from the correlation analysis. The authors are encouraged to use high speed cameras as in Ozaki et al. (2018). They reported high correlation of fast modulation of pulsating aurora. https://agupubs.onlinelibrary.wiley.com/doi/full/10.1029/2018GL079812

Line 294 It appears to be cloudy or overcast in Event 3. The imager data doesn't show clear pulsating auroral patches and is not appropriate for the correlation analysis. This event should be removed from the event list.

The event list should be re-created by considering the comments above and the conclusion of low correlation should be re-examined.

Line 210-215 The time scale of PA is discussed but the time scales of APA and PPA

are not. The degree of scattering and structure is labelled in the Figure 4 but it's very subjective and lacks quantitative definition. By looking at the plots in Figure 4, I don't see how the authors differentiated "less scatter, more structured" for PPA and "least scatter, most structured" for PA. Please give quantitative definition.

Line 232 It's problematic that waves in THEMIS FFT data outside the dusk sector are assumed to be all chorus. Hiss occurs in a wide area including the dawnside and dayside [Meredith et al., 2004, Shi et al., 2019]. The Meredith paper is cited but the authors only removed duskside hiss. The high occurrence on the dayside and dawnside in Figure 6 is most likely contaminated by hiss. Burst mode data should be used to differentiate chorus and hiss. https://agupubs.onlinelibrary.wiley.com/doi/epdf/10.1029/2004JA010387 https://agupubs.onlinelibrary.wiley.com/doi/full/10.1029/2018JA026041

Section 2 and 3 are logically disconnected from Section 4 and 5. Section 2 and 3 conclude that it's difficult to use satellite-aurora conjunctions to find many events with a good correlation between chorus and pulsating aurora. Then the authors gave up on using the conjunctions and used satellite and aurora data separately to continue the analysis in Section 5. The key conclusion comes only from Section 5 and not from Section 2 or 3. The authors claim that their approach avoids the issue of the low correlation but in that case Section 2 and 3 aren't really necessary. This logical disconnection makes it difficult to understand the story of this paper. It would make more sense to focus on Section 4 and 5 by removing Section 2 and 3.

Minor comments

The first three paragraphs in the introduction is too general and unfocused. This is not a textbook and there is no need to introduce the pitch angle, loss cone etc. These paragraphs should be removed and the introduction should focus on the topic of this paper.

Figure 7 doesn't give a fair comparison between the waves and aurora. PA is marked

where more than one events occur. But the waves are marked using a different threshold (>0.001 occurrence rate). The same threshold should be used. The different thresholds could also be the reason of the discrepancy discussed at line 306-310.

Line 314 It's unclear why the authors compare APA and long-lived chorus. Because they have different periods, there is no surprise that they are not related. APA should be compared to chorus with lifetime similar to APA.

---

## Author Comment (AC1) · 11 Feb 2020

We are writing to communicate our decision to withdraw this manuscript, "Inferring the source regions of pulsating auroras" (angeo-2019-162). We agree with many of the reviewers' concerns, and especially want to get the points that we are trying to make in sections 2 and 3 correct. It is our intention to split this paper into two separate publications and resubmit, starting first with a paper based on an enhanced view of the section 2 and 3 material. We thank the reviewers for their work.

Sincerely,
Eric Grono and Eric Donovan
* * *
[Figure]

2020.

ANGEOD

Interactive
comment